# Progress in Serial Imaging for Prognostic Stratification of Lung Cancer Patients Receiving Immunotherapy: A Systematic Review and Meta-Analysis

**DOI:** 10.3390/cancers16030615

**Published:** 2024-01-31

**Authors:** Hwa-Yen Chiu, Ting-Wei Wang, Ming-Sheng Hsu, Heng-Shen Chao, Chien-Yi Liao, Chia-Feng Lu, Yu-Te Wu, Yuh-Ming Chen

**Affiliations:** 1School of Medicine, National Yang Ming Chiao Tung University, Taipei 112, Taiwan; chiuhwayen@gmail.com (H.-Y.C.); eltonwang1@gmail.com (T.-W.W.); sam090904sam@gmail.com (M.-S.H.); hschao2@vghtpe.gov.tw (H.-S.C.); 2Institute of Biophotonics, National Yang Ming Chiao Tung University, Taipei 112, Taiwan; 3Department of Internal Medicine, Taipei Veterans General Hospital, Hsinchu Branch, Chutong 310, Taiwan; 4Department of Chest Medicine, Taipei Veterans General Hospital, Taipei 112, Taiwan; 5Institute of Biomedical Informatics, National Yang Ming Chiao Tung University, Taipei 112, Taiwan; 6Department of Biomedical Imaging and Radiological Sciences, National Yang Ming Chiao Tung University, Taipei 112, Taiwan; wl03151783@gmail.com (C.-Y.L.); elvin4016@ym.edu.tw (C.-F.L.)

**Keywords:** non-small cell lung cancer, radiomics, immunotherapy, immune checkpoint inhibitor, treatment outcome, computed tomography

## Abstract

**Simple Summary:**

Immunotherapy with checkpoint inhibitors is a promising treatment for lung cancer patients. However, not all patients respond well to immunotherapy, and researchers are seeking new predictive biomarkers for immunotherapy. Radiomics and its derivative, delta radiomics, are potential candidates for use as predictive biomarkers for use in immunotherapy. In this meta-analysis, we performed qualitative and quantitative assessments and confirmed the effectiveness of delta radiomics in predicting the treatment responses and clinical outcomes of immunotherapy. Further studies are warranted to compare the performance of traditional radiomics and deep-learning radiomics.

**Abstract:**

Immunotherapy, particularly with checkpoint inhibitors, has revolutionized non-small cell lung cancer treatment. Enhancing the selection of potential responders is crucial, and researchers are exploring predictive biomarkers. Delta radiomics, a derivative of radiomics, holds promise in this regard. For this study, a meta-analysis was conducted that adhered to PRISMA guidelines, searching PubMed, Embase, Web of Science, and the Cochrane Library for studies on the use of delta radiomics in stratifying lung cancer patients receiving immunotherapy. Out of 223 initially collected studies, 10 were included for qualitative synthesis. Stratifying patients using radiomic models, the pooled analysis reveals a predictive power with an area under the curve of 0.81 (95% CI 0.76–0.86, *p* < 0.001) for 6-month response, a pooled hazard ratio of 4.77 (95% CI 2.70–8.43, *p* < 0.001) for progression-free survival, and 2.15 (95% CI 1.73–2.66, *p* < 0.001) for overall survival at 6 months. Radiomics emerges as a potential prognostic predictor for lung cancer, but further research is needed to compare traditional radiomics and deep-learning radiomics.

## 1. Introduction

### 1.1. Overview of Lung Cancer and Its Global Burden

Lung cancer, a complex and heterogeneous disease, has long been recognized as a major global health issue owing to its high incidence and mortality rate [1]. Lung cancer is the leading cause of cancer-related deaths and is responsible for over 1.8 million fatalities annually, accounting for 18% of all cancer-related deaths worldwide [2]. The disease is primarily classified into two histological subtypes: non-small cell lung cancer (NSCLC), which represents approximately 85% of all cases, and small cell lung cancer (SCLC), which constitutes the remaining 15% [3]. The primary risk factors for lung cancer are tobacco smoking, exposure to environmental carcinogens such as radon gas and asbestos, air pollution, and genetic predispositions [4,5]. Although recent advancements in early detection and therapeutic interventions have led to improved clinical outcomes for some patients, the overall five-year survival rate for lung cancer remains low, at approximately 18% [6,7]. This highlights the urgent need for innovative diagnostic and prognostic tools to optimize treatment outcomes and tailor personalized therapy strategies for patients receiving advanced treatments, such as immunotherapy [8,9].

### 1.2. Role of Immunotherapy in Lung Cancer Treatment

The advent of immunotherapy has revolutionized the landscape of lung cancer treatment, offering new therapeutic options for patients and significantly affecting clinical outcomes [10]. Immunotherapy primarily focuses on recovering the power of the immune system to recognize and eliminate cancer cells, thereby providing a targeted and personalized approach to treatment [11]. Recently, immune checkpoint inhibitors (ICIs), which modulate the immune system by targeting immune checkpoints, such as programmed cell death protein 1 (PD-1) [12,13], programmed cell death ligand 1 (PD-L1) [14], and cytotoxic T-lymphocyte-associated protein 4 (CTLA-4) [15], have been enrolled in the standard of care for both NSCLC and SCLC [14,16]. These novel agents have demonstrated improved response rates, prolonged progression-free survival (PFS), and enhanced overall survival (OS) compared with conventional chemotherapy in several clinical trials [17,18]. Despite these promising results, not all patients respond to immunotherapy, and some may experience immune-related adverse events (irAEs) [19]. Consequently, ongoing research seeks to identify predictive biomarkers and develop combinatorial strategies to maximize treatment efficacy while minimizing adverse effects [20].

### 1.3. Importance of Delta Radiomics in Predicting Treatment Outcomes

In recent years, radiomics has emerged as a promising tool for predicting treatment outcomes in cancer patients, including those with lung cancer [21]. Radiomics is a quantitative image analysis approach that extracts high-dimensional, mineable data from medical images, such as computed tomography (CT) scans, magnetic resonance imaging (MRI), and positron emission tomography (PET) scans [22]. Delta radiomics, a subtype of radiomics, specifically focuses on changes in radiomics features over time, capturing the temporal evolution of the tumor phenotype and treatment response [23]. By providing a noninvasive and comprehensive analysis of tumor heterogeneity and progression, delta radiomics has the potential to significantly impact clinical decision making in personalized medicine, particularly in the era of immunotherapy [8,24]. Furthermore, delta radiomics may aid in identifying patients who are likely to respond to specific treatments, monitoring treatment efficacy, and predicting the risk of disease recurrence and metastasis [25]. Ongoing research aims to optimize the clinical application of delta radiomics by validating its predictive accuracy and addressing the methodological challenges related to reproducibility and standardization proposed by the image biomarker standardization initiative (IBSI) [26].

### 1.4. Objectives and Hypothesis of the Meta-Analysis

In recent years, there has been a significant surge in research aimed at elucidating the clinical implications of delta radiomic features obtained from computed tomography (CT) images in patients with NSCLC. These studies highlight the association between delta radiomic features and treatment response or clinical outcomes following immunotherapy. The primary objective of this investigation was to conduct an exhaustive systematic review of the existing body of delta radiomic research, particularly focusing on its potential to predict treatment response or outcomes in patients with NSCLC undergoing immunotherapy. This review entailed an assessment of the methodological quality of delta radiomic studies, utilizing the Quality in Prognosis Studies (QUIPS [27]) tool for image-mining research and the radiomics quality scoring (RQS [28]) tool as reference standards. Moreover, quantitative analysis was performed to determine the effectiveness of delta radiomics in predicting treatment response and clinical outcomes of immunotherapy in this cohort of patients.

## 2. Materials and Methods

### 2.1. Search Strategy and Selection Criteria

#### 2.1.1. Databases and Search Terms

This systematic review and meta-analysis were conducted according to the PRISMA guidelines [29]. The study has not been registered. To ensure meticulous adherence to these guidelines, checklists corresponding to PRISMA were utilized and are presented in Appendix A. A comprehensive literature search was performed using the following electronic databases: PubMed, Embase, Web of Science, and the Cochrane Library. Articles from 2017 January 1st to 2023 April 4th were searched. The search strategy comprised a combination of MeSH terms and keywords pertinent to lung cancer, immunotherapy, and delta radiomics. Example search terms include “lung cancer”, “non-small cell lung cancer”, “NSCLC”, “immunotherapy”, “PD-1”, “PD-L1”, “nivolumab”, “pembrolizumab”, “immunomodulating agent”, “radiomics”, “radiomic features”, “radiomic signature”, “texture analysis”, “delta radiomics”, “follow-up CT”, “serial imaging”, “early response”, “treatment monitoring”, “longitudinal monitoring”, “machine learning”, “deep learning”, “PFS”, “progression-free survival”, “survival”, “overall survival”, “OS”, “treatment outcome”, “response”, and “prediction”. A tailored search strategy was devised for each database to ensure an exhaustive and systematic review of the available literature.

#### 2.1.2. Inclusion and Exclusion Criteria

Studies were considered for inclusion if they fulfill the following criteria: (1) original research articles encompassing retrospective and prospective studies; (2) focus on patients with lung cancer treated with immunotherapy; (3) evaluate the prognostic significance of delta radiomics using CT scans to predict treatment outcomes; (4) report pertinent statistical measures, such as hazard ratios or odds ratios, for survival or other clinical endpoints; and (5) are published in English. The exclusion criteria comprise the following: (1) review articles, conference abstracts, case reports, editorials, and supplementary materials; (2) studies with insufficient data or ambiguous methodology; and (3) studies not centered on the application of delta radiomics in patients with lung cancer undergoing immunotherapy. Two independent reviewers (T.-W.W. and M.-S.H.) screened the titles. After reading the abstracts, a full-text evaluation was carried out to determine the eligibility for inclusion. Discrepancies between the reviewers were sent to a third reviewer (H.-Y.C.).

### 2.2. Data Extraction and Quality Assessment

#### 2.2.1. Data Extraction Process

Two independent reviewers (T.-W.W. and M.-S.H.) extracted data from the included studies using a standardized data extraction form (Appendix A), ensuring an accurate and comprehensive collection of information. The extracted information includes the following: (a) study characteristics such as authors, year of publication, study duration, the country where the study was conducted, study design, and sample size; (b) patient demographics, including age, sex, and stage of lung cancer; (c) delta radiomic features, encompassing preprocessing, imaging modality, software, segmentations, feature extraction methodologies, radiomic signature, and formula; (d) clinical and molecular data, including smoking history, type of immunotherapy, biomarkers, and other relevant clinical factors; and (e) outcome measures, namely area under the curve (AUC) of 6-month response, the hazard ratio of PFS, and OS. Discrepancies between reviewers were resolved through discussion or, if required, consultation with a third reviewer to ensure a consistent and rigorous analysis.

#### 2.2.2. Quality Assessment

The methodological quality of the studies included in this review was meticulously evaluated using two assessment tools: the QUIPS [27] tool and the RQS [28] tool. The QUIPS tool, specifically designed for prognostic studies, examines the risk of bias and applicability concerns across six domains: participant selection, study attrition, measurement of prognostic factors, assessment of outcomes, control of confounding factors, and statistical analysis, and reporting procedures [27]. Bias risk was assessed for all six domains, while applicability concerns were addressed for the initial three domains. In contrast, the RQS tool, developed to assess the validity and potential bias of radiomics studies, consists of 16 components [8]. Each study was assigned a score for every RQS component, and these scores were then aggregated to generate a total score. To ensure consistency and accuracy in the quality assessment process, two independent reviewers performed evaluations. If any discrepancies arose between the reviewers, they engaged in discussion to reach a consensus or, if necessary, seek input from a third reviewer.

### 2.3. Definitions of 6-Month Progression-Free Survival and Overall Survival

The definitions of 6-month PFS and treatment response at the 6th month were interchangeable among studies; thus, we collectively defined them as 6-month response: whether the disease progressed within 6 months after treatment. The 6-month OS was defined as whether the patient was alive within 6 months after treatment.

### 2.4. Meta-Analysis

We conducted 3 distinct meta-analyses using the studies we included. (1) The first was a meta-analysis evaluating the predictive performance of delta radiomics models for 6-month response to immunotherapy using a fixed-effects model. We used AUC as the performance evaluation metric. The AUC, derived from the receiver operating characteristic (ROC) curve, represents a graphical depiction of a model’s sensitivity against its false-positive rate (1–specificity) at different threshold settings. When the AUC has a value of 1, it indicates that the classification model performed perfectly, and a value of 0.5 indicates that the model is no better than random chance. When multiple AUC values were reported in a single study, we chose the best-performing model incorporating delta radiomics features for further inclusion in the meta-analysis. The AUC values were treated as expected values for further analysis. Additional subgroup analyses were performed according to traditional radiomics or deep-learning radiomics. (2) The other 2 meta-analyses compared the PFS and OS of immunotherapy between high- and low-risk groups in the validation datasets. We measured the performance by using the pooled hazard ratio (HR) and a 95% confidence interval (CI) using a fixed-effects model. The hazard ratio was transformed to a logarithmic scale. The 95% CI was used to back-calculate the standard deviation (SD) with the corresponding T-score from a Student’s T-distribution with n−1 degrees of freedom. In instances where a single study did not provide a 95% CI or SD but instead reported a standard error of the mean (SE), the SD was calculated by multiplying the square root of the sample size by the SE.

### 2.5. Statistical Analysis

Heterogeneity assessment among studies was conducted using Cochran’s Q test, and the I^2^ statistic was employed for quantification purposes. The I^2^ statistic gauges the proportion of variability in effect estimates resulting from heterogeneity as opposed to sampling error. I^2^ values of 25%, 50%, and 75% corresponded to low, moderate, and high heterogeneity, respectively. A random-effects model (DerSimonian–Laird method) was applied for the meta-analysis in the presence of significant heterogeneity (I^2^ > 50%). In contrast, a fixed-effects model (the Mantel–Haenszel method) was adopted when no significant heterogeneity was observed. Combined effects were calculated, and a two-sided *p* value of 0.05 was considered indicative of statistical significance [30]. Publication bias assessment was performed when more than 10 studies were included, as detecting funnel plot asymmetry requires a minimum of 10 studies [31]. All analyses were carried out using Review Manager (RevMan) [a computer program], version 5.4, Cochrane Collaboration, Paris, France, 2020.

## 3. Results

### 3.1. Study Selection and Characteristics

#### 3.1.1. Flow Diagram of Study Selection

Figure 1 depicts the flow diagram that outlines the study selection process utilized in this systematic review and meta-analysis. The initial search across PubMed (*n* = 48), Embase (*n* = 108), and Web of Science (*n* = 67) generated a total of 223 studies. Following the removal of duplicates, 176 studies remained for further evaluation. The titles and abstracts of the mentioned articles underwent screening, leading to the exclusion of 151 articles. Subsequently, 25 studies were evaluated for eligibility, resulting in the exclusion of 15 articles. Seven of these articles were excluded due to their study designs not being related to the research interest. Two articles were excluded as their outcomes were not pertinent to the focus of this review. Additionally, one article was a conference paper, one was a meeting abstract, and four articles were supplementary materials, all of which were not considered for inclusion in this systematic review and meta-analysis. In the end, 10 studies met the inclusion criteria and were incorporated into this systematic review and meta-analysis [32,33,34,35,36,37,38,39,40,41].

#### 3.1.2. Characteristics of Included Studies

The detailed characteristics of the 10 eligible studies are summarized in Table 1. The 10 studies enrolled a total of 1513 patients with advanced NSCLC treated with immunotherapy [32,33,34,35,36,37,38,39,40,41]. The basic characteristics of the studies are summarized in Table 1 and Table 2. Two studies were conducted in Spain [32,37], one in Belgium [33], four in China [33,35,36,38], one in the Netherlands [37] and two in the US [40,41]. All of the 10 studies were retrospective. The size of the study cohort ranges from 88 to 224. The median patient age ranged from 61 to 65 years, and the proportion of patients who were female ranged from 9% to 67%. CT was performed before and after immunotherapy treatment. The immunotherapy agent included anti-PD-1 and anti-PD-L1 as monotherapy or combination therapy with the other regimen.

#### 3.1.3. Radiomics and Image Analysis

The details of the radiomics, type of classifier, and included clinical features are presented in Table 2. To obtain radiomics, tumor segmentation was performed manually in nine studies [32,33,34,35,37,38,39,40,41]. One study calculated radiomics not from segmented tumors but from a whole-lung image [37]. For the studies with segmented tumors, primary tumors alone were segmented in seven studies [32,33,35,36,38,39,41], while the others defined regions of interest [34,40]. The included radiomic parameters are listed in Appendix A. Six studies disclosed the predictive radiomic parameters [33,34,35,38,40,41], while the other four studies did not describe the details of the radiomic parameters [32,36,37,39]. Although the names of the radiomic parameters and their biological interpretations differed in each study, both size-based radiomic features [33,38] and texture-based radiomic features [33,34,35,38,40,41] are significant predictive parameters. Clinical features were incorporated into five studies [32,33,34,36,38]. A variety of formulas were used to calculate the difference in radiomic features across studies, incorporating both follow-up and pretreatment information. The validation methods varied, with four studies using external testing [28,30,34,35] and others employing cross-validation or split-sample techniques. The classifier methods included random forest (RF) [32,33,36,40], LASSO-Cox [34,38,39], SVM [35], and LDA [40]. The endpoints assessed in these studies were predominantly PFS, OS, and treatment response [32,33,34,35,36,37,38,39,40,41].

### 3.2. Quality Assessment Results

The Quality in Prognosis Studies (QUIPS [27]) quality assessment of the 10 studies is shown in Figure 2 [32,33,34,35,36,37,38,39,40,41]. The assessment of the individuals showed that there was a low risk of bias and fair application concerns for most of the assessed criteria, except for a higher risk of study participants in two studies and confounding measurement in four studies (Figure 2a). A summary of the risk of bias for all studies is shown in Figure 2b.

Table 3 presents the details of radiomic quality assessment in six domains. The mean RQS [28] for the 10 studies was 13.5 (range: 10–26). A majority of the studies provided well-documented image acquisition protocols. Four studies fulfilled the multiple segmentation criteria with different methods. Dong X. et al. segmented both the tumor and region of interest [34]. Yi Y. et al. segmented the tumor with two readers [36]. Benito F. et al. segmented the tumor using an algorithm under the supervision of experienced radiologists [39]. Laurent D. et al. segmented the tumor using an algorithm, and the results were checked by a trained chest radiologist [41]. A phantom study was not used in all studies. Three studies acquired imaging at multiple time points [32,33,34]. All studies used feature reduction methods and validation datasets. Eight of the ten studies integrated non-radiomic features into the prediction models. Only Laurent D. et al. discussed the biological correlates of phenotype difference in terms of radiomics [41]. A comparison to the ‘gold standard’ was not performed in all studies. Most of the studies demonstrated the potential clinical utility of radiomic models. Six out of ten studies performed cut-off analysis. All studies reported discrimination statistics for further analyses. Only three studies reported calibration statistics for further resampling. Only Laurent et al. prospectively collected the retrospective data with registration in a trial database [41]. Stefano T. et al. performed a cost-effectiveness analysis and featured open science and data [37]. 

### 3.3. Delta Radiomic Features and Prognostic Performance

The patients could be stratified into low- and high-risk groups through the use of radiomic models. The first meta-analysis evaluates the predictive power of the prognosis model at 6-month response with AUC, showing that the pooled AUC was 0.81 (95% CI 0.76–0.86, *p* < 0.001) for 6-month response (six studies, Figure 3a). The I^2^ statistic implied low heterogeneity among the studies (I^2^ = 15.1%, *p* = 0.58). The traditional radiomics subgroup showed a higher pooled AUC of 0.84 (95% CI 0.77–0.92, *p* < 0.001) with lower heterogeneity among the studies (I^2^ = 0%); however, the deep-learning radiomics subgroup showed a lower pooled AUC of 0.79 (95% CI 0.72–0.86, *p* < 0.001) with higher heterogeneity among studies (I^2^ = 33%, *p* = 0.22).

The second and third meta-analyses of comparisons of the immunotherapy outcomes between the two groups showed that the pooled HR was 4.77 (95% CI 2.70–8.43, *p* < 0.001) for PFS (four studies; Figure 3b) and 2.15 (95% CI 1.73–2.66, *p* < 0.001) for OS (five studies; Figure 3c). The I^2^ statistic implied moderate and low heterogeneity among the studies (I^2^ = 58%, *p* = 0.07; I^2^ = 48%, *p* = 0.10). Funnel plots of PFS AUC, PFS HR, and OS HR are presented in Figure 4. The Egger’s tests for all endpoints showed a low risk of publication bias (*p* = 0.1148 for PFS AUC, *p* = 0.1198 for PFS HR, *p* = 0.0885 for OS HR).

## 4. Discussion

### 4.1. Summary of Key Findings

We conducted this meta-analysis to assess the prognostic ability of delta radiomics in immunotherapy in NSCLCs. In this meta-analysis encompassing 10 studies, incorporated into a qualitative synthesis, we integrated 6 studies to assess AI performance in predicting the 6-month response to immunotherapy through serial imaging, 4 studies to forecast the hazard ratio for PFS, and 5 studies for the hazard ratio for OS. All studies were retrospective in nature. Some studies showed a high risk of bias in study participation and study confounding. The RQS was 13.5 (range: 10–26) out of 36, indicating a moderate level of quality. The predictive parameters varied in terms of both radiomic formulas and the inclusion of clinical features. Despite variations in materials and AI models among the studies, all AI models effectively categorized patients undergoing immunotherapy into low- and high-risk groups across all three aspects: 6-month response, PFS hazard ratio, and OS hazard ratio.

### 4.2. Comparison with Previous Studies and Literature

To the best of our knowledge, this is the first meta-analysis evaluating the application of delta radiomics in prognosis stratification in lung cancer treated with ICIs. Two previous meta-analyses [43,44] have demonstrated the role of radiomics in predicting lung cancer prognosis. Our meta-analysis further extends the landscape to delta radiomics and multi-omic models. Compared to our previous meta-analysis about the application of radiomics in prognosing lung cancer treated with epidermal growth factor receptor tyrosine kinase inhibitors (EGFR-TKI) [45], the quality of the studies about immunotherapy was better, except in the study-confounding domain. The mean RQS was 13.5 in studies about immunotherapy and 11.67 in studies about EGFR-TKI, both indicating a moderate level of quality. In another systemic review about tracking tumor biology with radiomics [28], most of the studies (30 out of 41) included performed at less than 12 out of 36 (<30%) in terms of RQS. The mean RQS was 10.66 (29.6%) in studies about predicting immunotherapy response in NSCLC [43]. The RQS in all meta-analyses mentioned previously indicated a moderate level of quality. Most studies missed the quality in “Phantom Study on All Scanner”, “Detect and Discuss Biological Correlates”, “Comparison to ‘Gold Standard’”, “Prospective Study Registered in a Trial Database”, “Cost-Effectiveness Analysis”, and “Open Science and Data”. With the goal of discovering new biomarkers and developing a prediction model, some RQS domains may not be applicable. However, it would be important to discuss the biological correlates to make biomarkers interpretable. Open science and data is another important domain, and by combining discrimination statistics and calibration statistics, researchers could pile all the data together to train better models and perform more analyses.

### 4.3. Comparison of Radiomic Features in the Enrolled Studies

While the 10 enrolled studies utilized radiomics/delta radiomics as predictive parameters, the radiomic features employed in the predictive models differ across these studies. To better understand the biological interpretation of radiomic features, we categorized radiomics into several groups based on IBSI [26]. Among the six studies that disclosed their predictive radiomic features, a majority of these features are texture-related, such as entropy [33,35,41], contrast [41], and grey-level uniformity [34]. The findings regarding the importance of texture-based radiomics align with the heterogeneity observed in tumors, a factor proven to be associated with prognosis in solid tumors [46,47]. Only in two studies did size-based radiomic features, such as tumor volume [33] and axis length [38], enter the final prediction model. These findings suggest that changes in texture may provide more information than the traditional volume change proposed by the Response Evaluation Criteria in Solid Tumors (RECIST) working group or its modified version for immunotherapy, iRECIST [48]. Furthermore, radiomics that captured both size and heterogeneity were also deemed significant. For instance, metrics such as large/small area and low/high grey-level emphasis [34,35,38], which address both size and heterogeneity, were employed in three studies included in our analysis.

### 4.4. Strength and Limitations

The most significant strength of this study is its status as the first meta-analysis discussing the predictive power of delta radiomics. Both QUIPS and RQS assessments were conducted in this study. However, several limitations were encountered. Firstly, out of the 223 studies/articles searched, only 10 studies could be included in the final analysis, indicating that delta radiomics is a relatively novel predictive parameter. Secondly, limited data hindered our ability to perform pooled sensitivity, pooled specificity, and heterogeneity assessments as well as threshold effect assessment (HSROC). Therefore, we present the funnel plots in Figure 4, which demonstrate a low risk of publication bias. Thirdly, the varying definitions of radiomics and the inclusion of different radiomic features in various publications made it challenging for us to conduct further analysis. Finally, the immunotherapy agents varied across different studies, and detailed information could not be obtained from some studies. This limitation prevents us from evaluating the individual effects of immunotherapy.

### 4.5. Future Directions and Research Opportunities

To extend the application of radiomics and delta radiomics for predicting prognoses of NSCLC patients receiving immunotherapy with checkpoint inhibitors, future studies should focus on the subsections below.

#### 4.5.1. External Validation

As shown in our PRISMA evaluation, nearly half of the studies are at high risk of bias in study confounding. Using external validation and standardized protocols could ensure the reliability and reproducibility of predicting models [49]. The validation of the model with either data from other hospitals or temporal data is applicable [50].

#### 4.5.2. Biomarker Interpretation

In our systemic review, a limited number of studies have interpreted the selected radiomic parameters in prediction models, echoing similar findings in other meta-analyses [43,44,45] focusing on radiomics predicting lung cancer prognosis. Despite radiomics being derived from images, understanding the significance of calculations is crucial. As the field progresses into deep-learning parameters, the challenge shifts to providing interpretative tools like heat maps [51,52] for a comprehensive understanding.

#### 4.5.3. Trial Registration and Quality Insurance

As the convenience of employing supercomputers for model development with ample data increases, guarding against overfitting becomes pivotal. A registered study design prior to analysis serves as a preventive measure against overfitting. Additionally, the utilization of tools like the QUIPS tool [27], the QUADUS tool [53], and the RQS tool [28] is indispensable, serving as reminders to authors about potential pitfalls and safeguarding against the risk of bias.

#### 4.5.4. Data Sharing

A notable limitation in our systematic review and meta-analysis is that, although 10 studies entered the analysis phase, only 3 to 5 studies could be included in each meta-analysis due to a lack of discrimination/calibration statistics or missing study endpoints. This limitation could potentially be addressed with a standardized data-sharing format. As more parameters and drugs are continually developed, leveraging previous knowledge becomes crucial for scientific progress. Encouragingly, both geographic and temporal dimensions of data sharing are vital. We advocate for future authors to engage in open databases such as The Cancer Imaging Archive [54], ensuring compliance with institute review board standards and adherence to Findable–Accessible–Interoperable–Reusable (FAIR) data principles. For model sharing, federated learning emerges as a potential breakthrough in the future. Compliance to the institute review board and Findable–Accessible–Interoperable–Reusable (FAIR) data principles [55] could be a solution. For model sharing, federated learning could be a future breakpoint.

## 5. Conclusions

Our systemic review and meta-analysis demonstrated the usefulness of delta radiomics as a biomarker to predict the prognosis of NSCLC patients receiving immunotherapy with checkpoint inhibitors. With models containing delta radiomics, the performance was 0.81 in terms of the AUC when predicting 6-month response, and the stratified hazard ratio of the high-risk/low-risk groups was 4.77 for PFS and 2.15 for OS. To further optimize the ability of the models in clinical settings, future authors could register the trial before starting, focus on external validation and biomarker interpretation in study design, and share the data in a standard format for further analysis.

## Figures and Tables

**Figure 1 cancers-16-00615-f001:**
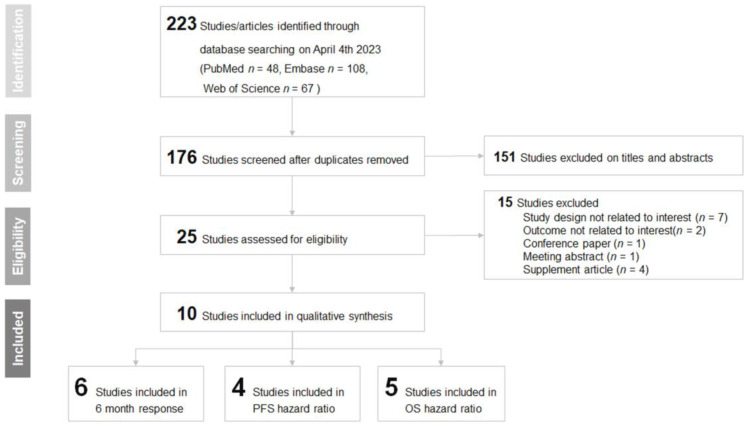
PRISMA flowchart of the included studies.

**Figure 2 cancers-16-00615-f002:**
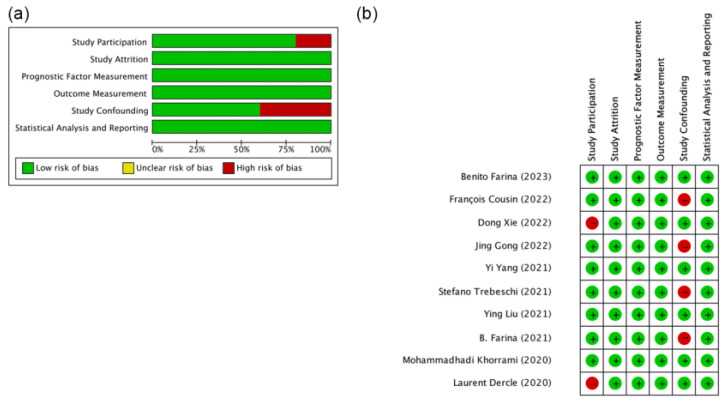
Quality assessment. (**a**) Risk of bias for individual studies. (**b**) Summary of risk of biases [32,33,34,35,36,37,38,39,40,41].

**Figure 3 cancers-16-00615-f003:**
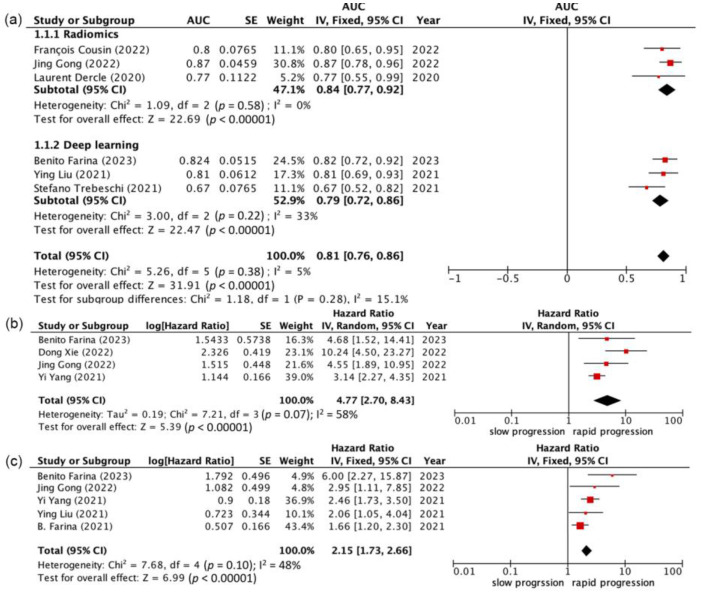
Forest plots of the predictive performance of radiomics models in progression-free survival and overall survival of NSCLC patients treated with immunotherapy [32,33,34,35,36,37,38,39,41]. (**a**) The 6-month response, (**b**) hazard ratio for progression-free survival, (**c**) hazard ratio for overall survival.

**Figure 4 cancers-16-00615-f004:**
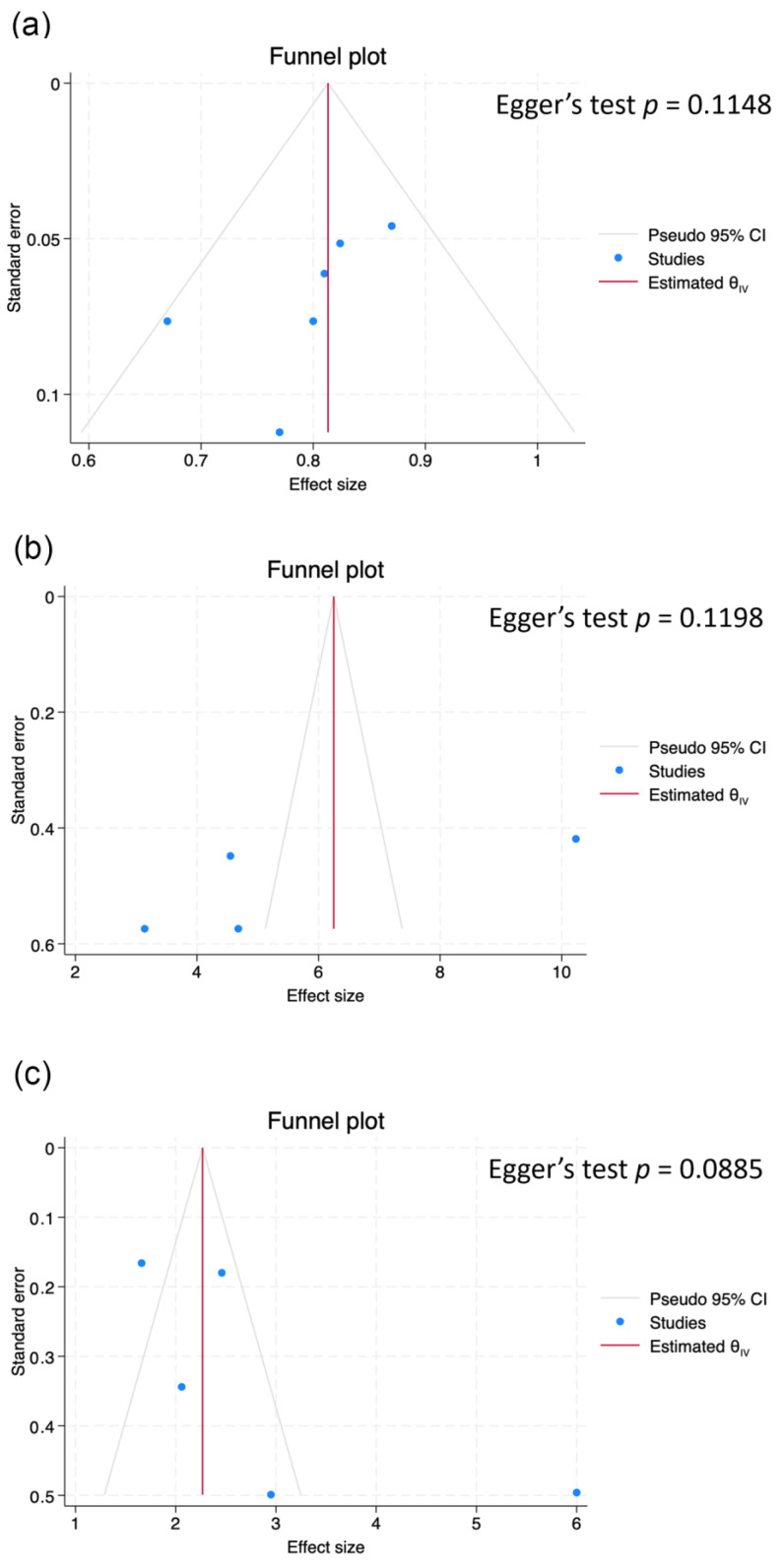
Funnel plots of the pooled predictive performance of radiomics models in progression-free survival and overall survival of NSCLC patients treated with immunotherapy. (**a**) The 6-month response AUC, (**b**) hazard ratio for progression-free survival, (**c**) hazard ratio for overall survival.

**Table 1 cancers-16-00615-t001:** Basic characteristics of studies included in systematic review and the meta-analysis.

Author	Dataset	Study Duration	Country	Study Design	Patients	Age	Sex (Female)	Smoker	Stage	Adeno	Immunotherapy Agent	Immunotherapy Regimen
Benito, F. (2023) [32]	D	2013–2021	Spain	Retrospective	200	65 (58–70)	58 (29)	174 (87.9)	IV	151 (75.5)	Immunotherapy	Monotherapy or combination
François, C. (2022) [33]	D	2015–2020	Belgium	Retrospective	121	65 (41– 85)	45 (37)	119 (98)	III–IV	71 (59)	Pembrolizumab, nivolumab, atezolizumab	Monotherapy
	E	2015–2018	Belgium	Retrospective	39	64 (44– 95)	19 (49)	32 (82)	III–IV	29 (74)	Pembrolizumab, nivolumab, atezolizumab	Monotherapy
Dong, X. (2022) [34]	D	2016–2021	China	Retrospective	68	NA	6	35	III–IV	23	Camrelizumab, sintilimab, tislelizumab, nivolumab, atezolizumab	Monotherapy or combination
	V	2016–2021	China	Retrospective	29	NA	3	15	III–IV	6	Camrelizumab, sintilimab, tislelizumab, nivolumab, atezolizumab	Monotherapy or combination
Jing, G. (2022) [35]	D	2015–2018	China	Retrospective	93	67 (31–85)	13 (14)	42 (45.2)	III–IV	57 (61.3)	Immunotherapy	Monotherapy
	E	2016–2020	China	Retrospective	68	61 (27–76)	16 (23.5)	46 (67.6)	III–IV	54 (79.4)	Immunotherapy	Monotherapy
	E	2018–2020	China	Retrospective	63	66 (29–86)	11 (17.5)	19 (30.2)	III–IV	38 (60.3)	Immunotherapy	Monotherapy
Yi, Y. (2021) [36]	D	2016–2019	China	Retrospective	200	NA	35	119	IIIB–IV	132 (66)	Nivolumab, pembrolizumab, atezolizumab	Monotherapy
Stefano, T. (2021) [37]	D	2014–2016	Netherlands	Retrospective	152	64.4 (57.8–68.9)	64	x	IV	92 (60.5)	Anti-PD1 immunotherapy	Monotherapy
Ying, L. (2021) [38]	D	2018–2019	China	Retrospective	112	NA	14	81	NA	61	Immunotherapy	Monotherapy or combination
	V	2018–2019	China	Retrospective	49	NA	13	35	NA	23	Immunotherapy	Monotherapy or combination
Benito, F. (2021) [39]	D	2013–2019	Spain	Retrospective	88	NA	NA	NA	NA	NA	Immunotherapy	Monotherapy or combination
Mohammadhadi, K. (2020) [40]	D	2012–2017	America	Retrospective	112	65 (42–83)	54 (48)	96 (86)	NA	80 (71)	Nivolumab, pembrolizumab, atezolizumab	Monotherapy
	E	2012–2017	America	Retrospective	27	63 (42–83)	18 (67)	21 (78)	NA	21 (78)	Nivolumab, pembrolizumab, atezolizumab	Monotherapy
Laurent, D. (2020) [41]	D	NA	America	Retrospective	92	NA	NA	NA	III–IV	0	Nivolumab	Monotherapy

D, development dataset; V, validation dataset; E, external validation dataset; NA, not applicable; Adeno, adenocarcinoma.

**Table 2 cancers-16-00615-t002:** Summary of details of radiomic and image analyses.

Author	Segmentation	VOI	Clinical Feature	Radiomics	Formula	Software	Validation	Classifier	EndPoints
Benito, F. (2023) [32]	Manual	primary tumor	NLR, SII, Hb, MLR, neutrophil, liver metastasis, histology, platelet, smoking, PLR, BMI, age	longitudinal radiomics	concatenate pretreat and follow	NoduleX (deep learning)	Cross validation	Random forest	PFS, OS
François, C. (2022) [33]	Manual	primary tumor	sex, clinical stage, ANC, eosinophil, and NLR	delta radiomics	follow-pretreat	Radiomics (Oncoradiomics SA, Belgium)	External testing	RF, CoxPH	Response, OS
Dong, X. (2022) [34]	Manual	tumor	tumor anatomical classification and brain metastasis	delta radiomics	follow-pretreat	Pyradiomics	Split sample	LASSO-Cox	PFS
Jing, G. (2022) [35]	Manual	primary tumor	NA	delta radiomics	follow-pretreat	Pyradiomics	External testing	SVM	Response, PFS, OS
Yi, Y. (2021) [36]	Manual	primary tumor	clinical + blood test	longitudinal radiomics	SimTa module	Pyradiomics	Cross validation	SimTA	Response, PFS, OS
Stefano, T. (2021) [37]	NA	whole lung	NA	longitudinal radiomics	deep feature	VGG-like network	Split sample	RF	PFS, OS
Ying, L. (2021) [38]	Manual	primary tumor	distant metastasis	delta radiomics	(follow-pretreat)/pretreat	Analysis Kit, version 3.2.5, GE Healthcare	Split sample	LASSO-Cox	Response
Benito, F. (2021) [39]	Manual	primary tumor	NA	delta radiomics	follow-pretreat	Pyradiomics	External testing	LASSO-Cox	OS
Mohammadhadi, K. (2020) [40]	Manual	tumor	NA	delta radiomics	follow-pretreat	In-house developed toolbox with MATLAB 2018b	External testing	LDA	Response, OS
Laurent, D. (2020) [41]	Manual	primary tumor	NA	delta radiomics	size: (follow-pretreat)/pretreat, other: follow-pretreat	In-house developed toolbox	Split sample	RF	Response

ANC, absolute neutrophil count; BMI, body mass index; LASSO-Cox, least absolute shrinkage and selection operator–Cox proportional hazards; LDA, linear discriminant analysis; MLR, monocyte to lymphocyte ratio; NA, not applicable; NLR, neutrophil to lymphocyte ratio; OS, overall survival; PFS, progression-free survival; PLR, platelet to lymphocyte ratio; RF, random forest; SII, systemic inflammation index; SVM, support vector machine; VOI, volume of interest.

**Table 3 cancers-16-00615-t003:** Details of radiomics quality scores.

	Domain 1				Domain 2		Domain 3				Domain 4			Domain 5		Domain 6	
Author	Image Protocol Quality	Multiple Segmentation	Phantom Study on All Scanner	Imaging at Multiple Time Points	Feature Reduction or Adjustment for Multiple Testing	Validation	Multivariable Analysis with Non -Radiomic Features	Detect and Discuss Biological Correlates	Comparison to ’Gold Standard’	Potential Clinical Utility	Cut-Off Analyses	Discrimination Statistics	Calibration Statistics	Prospective Study Registered in a Trial Database	Cost-Effectiveness Analysis	Open Science and Data	Total
Benito, F. (2023) [32]	1	0	0	1	3	3	1	0	0	1	0	2	0	0	0	0	12
François, C. (2022) [33]	1	0	0	1	3	3	1	0	0	1	0	2	0	0	0	0	12
Dong, X. (2022) [34]	1	1	0	1	3	2	1	0	0	1	1	2	1	0	0	0	14
Jing, G. (2022) [35]	0	0	0	0	3	3	0	0	0	1	1	2	0	0	0	0	10
Yi, Y. (2021) [42]	1	1	0	0	3	3	1	0	0	1	1	2	0	0	0	0	13
Stefano, T. (2021) [37]	1	0	0	0	3	3	1	0	0	1	0	2	0	0	1	1	12
Ying, L. (2021) [38]	1	0	0	0	3	3	1	0	0	1	0	2	1	0	0	0	12
Benito, F. (2021) [39]	1	1	0	0	3	3	0	0	0	0	1	2	0	0	0	0	11
Mohammadhadi, K. (2020) [40]	1	0	0	0	3	4	1	0	0	1	1	2	0	0	0	0	13
Laurent, D. (2020) [41]	1	1	0	0	3	2	1	1	0	2	1	2	2	7	0	0	26

## Data Availability

The data presented in this study are available upon request from the corresponding author.

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
