# Peer review of "Progress in Serial Imaging for Prognostic Stratification of Lung Cancer Patients Receiving Immunotherapy: A Systematic Review and Meta-Analysis"

_cancers, 2024, doi:10.3390/cancers16030615_

Round 1

Reviewer 1 Report

Comments and Suggestions for Authors

The topic of the study is currently really a hot topic. However, in this "mare magnum" of articles published on radiomics in the literature, a new and non-homegenised topic, confusion can arise. The authors submitted a meta-analysis with the aim of providing some clarity on such a delicate and important topic, trying to clarify if Radiomics can be a potential prognostic predictor for lung cancer. From 223 articles they were only able to include 10 articles in the meta-analysis and this is undoubtedly a limitation. However, the authors followed a strict exclusion criterion which excluded the majority of works and this confirms what was said about the heterogeneity of the articles dealing with this topic. The analysis method is good, as is the English language. I think it deserves publication

Author Response

We thank the reviewer's comments, which accurately summarized our article and the aim of our research.

Reviewer 2 Report

Comments and Suggestions for Authors

60-61, immunotherapy does not augment the capacity of the immune system as the proposal implies but unblocks its immunity caused by the specific receptors it targets

62-63 Immunotherapy is no longer promising for both SCLC and NSCLC, but is an established practice and standard of care.

The parameters that delta radiomics uses to infer changes that occur during imaging monitoring have not been discussed. In section 3.1.3. Radiomics and Image analysis, parameters used by the various studies selected are briefly described, but without a choice by the authors of the proposed methodology in the discussion.Please analyze

There is limited discussion on the biological interpretation of selected radiomic parameters, which is crucial for understanding the mechanistic underpinnings of the models' predictions

Author Response

60-61, immunotherapy does not augment the capacity of the immune system as the proposal implies but unblocks its immunity caused by the specific receptors it targets

==>The wordings were amended to "Immunotherapy primarily focuses on recovering the power of the immune system to recognize and eliminate cancer cells, ..."

62-63 Immunotherapy is no longer promising for both SCLC and NSCLC, but is an established practice and standard of care.

==>The wordings were amended to "..., have been enrolled in the standard of care for both NSCLC and SCLC."

The parameters that delta radiomics uses to infer changes that occur during imaging monitoring have not been discussed. In section 3.1.3. Radiomics and Image analysis, parameters used by the various studies selected are briefly described, but without a choice by the authors of the proposed methodology in the discussion. Please analyze

==>We searched the supplementary materials and made a list in the supplementary material. Because the parameters used in different studies vary a lot, and the names of the radiomic parameters are not in a regular form. We could only list them and tried our best to unify their biological meanings in results and discussion.
section 3.1.3. Radiomics and Image analysis: " The included radiomic parameters were listed in Table S1. Six studies disclosed the predictive radiomic parameters[33-35, 38, 40, 41], while the other 4 studies did not describe the details of radiomic parameters[32, 36, 37, 39]. Although the names of the radiomic parameters and their biological interpretations differed in each study, both size-based radiomic features[33, 38] and texture-based radiomic features[33-35, 38, 40, 41] are significant predictive parameters. "

There is limited discussion on the biological interpretation of selected radiomic parameters, which is crucial for understanding the mechanistic underpinnings of the models' predictions
==>We had listed the parameters in the supplementary material and added a paragraph to discuss it:
4.3. Comparison of radiomic features in the enrolled studies
While the 10 enrolled studies utilize radiomics/delta radiomics as predictive parameters, the radiomic features employed in the predictive models differ across these studies. To better understand the biological interpretation of radiomic features, we categorized radiomics into several groups based on IBSI[26]. Among the 6 studies that disclosed their predictive radiomic features, a majority of these features are texture-related, such as entropy[33, 35, 41], contrast[41], and grey level uniformity[34]. The findings regarding the importance of texture-based radiomics align with the heterogeneity observed in tumors, a factor proven to be associated with prognosis in solid tumors[46, 47]. Only in 2 studies did size-based radiomic features, such as tumor volume[33] and axis length[38], enter the final prediction model. These findings suggest that changes in texture may provide more information than the traditional volume change proposed by the Response Evaluation Criteria in Solid Tumors(RECIST) working group or its modified version for immunotherapy, iRECIST[48]. Furthermore, radiomics that captured both size and heterogeneity were also deemed significant. For instance, metrics such as large/small area low/high grey level emphasis[34, 35, 38], which address both size and heterogeneity, were employed in 3 studies included in our analysis.

Reviewer 3 Report

Comments and Suggestions for Authors

Cancers-2775797

Progress in Serial Imaging for Prognostic Stratification of Lung 2 Cancer Patients Receiving Immunotherapy: A Systematic review and Meta-Analysis.

The authors have investigated the role of delta radiomics in response assessment, and other clinical outcomes (PFS and OS) in lung cancer patients treated with immunotherapy through a systematic review and metaanalysis of only 10 studies. This the basic limitation of the manuscript

The authors have followed all the relavent PRISMA guidelines for conducting the metaanalysis and systematic review

Comments

1.       Although the authors are doing a metaanalysis of studies with delta radiomics as a predictive biomarker for response assessment to immunotherapy: the title does not show that. The authors are advised to have another title where delta radiomics is represented.

2.       The manuscript has lots of typographical errors and also errors in sentence construction.

       This needs to be corrected.

3.       In the section of standardised data extraction, the authors have mentioned about the standardised data extraction for. I would suggest them to put the detailed form in the supplementary Material as that will be useful in the QA of the metaanalysis/Systematic review.

4.       In the Search strategy, the authors need to put the time period from which year to which year they had searched for the articles that match their search criteria.

5.       For QA the authors used the QUIPS AND RQS tool: What was the scoring of both the tools

Secondly the authors need to put the filled-up tool for their study in the supplementary table as this needs to be reviewed.

6.       The have to define Overall survival also as they have defined response and PFS.

7.       In figure -1 Instead of records, the authors should mention studies/articles

8.       Heterogeneity Assessment Threshold Effect Assessment (HSROC) gives the sources of heterogeinity. I would recommend the authors to include a HSROC curve as it will add to the scientific rigour of the manuscript.

9.       In the results section, the authors have used only one diagnostic metrics i.e is AUC. I understand that AUC is important and the primary diagnostic metric in a metaanalysis. However other diagnostic metrics like pooled specificity, pooled sensitivity should be calculated and should be represented in a forest plot as done for AUC.

10.    In table -2 the coloumn having clinical features is redundant as it has not been used for the analysis purpose, hence the coloumn of clinical features can be removed from the table 2.

11.   The table 2 should include the radiomic features that were significant in each study for the given endpoints

12.   A funnels plot for PFS, OS should be plotted for publication bias depicting the asymmetry of the funnel plot. The authors should also apply the eggers test for statistical significance for any publication bias.

13.   Discussion section

a. The discussion should start with endpoint in question and why this metaanalysis / systematic review is being done.

b. The authors should discuss about the strength and limitations of the 10 studies.

c. If I am not mistaken, all the studies had different Immunotherapy agents which would actually change the response rates and the OS. The authors are encouraged to describe this in detail in the manuscript and should explain how it might affect their study.

c. The authors should describe in details, the strengths and the weakness of their study which is a very important aspect of the metaanalysis.

 d. since this manuscript deals with delta-radiomics, the authors are advised to also dwell on the radiomic features in each study in the result section and also to include it to expand the discussion section

Comments on the Quality of English Language

The quality of English needs to be improved

Author Response

Comments

1.       Although the authors are doing a metaanalysis of studies with delta radiomics as a predictive biomarker for response assessment to immunotherapy: the title does not show that. The authors are advised to have another title where delta radiomics is represented.

==> We considered "Serial Imaging" and "Delta Radiomics" for the title. However, as delta radiomics is derived from radiomics obtained through serial imaging, we opted for "Serial Imaging" because it is more user-friendly for researchers. "Delta Radiomics" might be less intuitive to understand.

2.       The manuscript has lots of typographical errors and also errors in sentence construction.

       This needs to be corrected.
==> Thanks for your advice. We corrected them.

3.       In the section of standardised data extraction, the authors have mentioned about the standardised data extraction for. I would suggest them to put the detailed form in the supplementary Material as that will be useful in the QA of the metaanalysis/Systematic review.
==> Thanks for the suggestion. We would put the form of standardized data extraction in the supplementary material.

4.       In the Search strategy, the authors need to put the time period from which year to which year they had searched for the articles that match their search criteria.
==> We searched articles from 2017/01/01 to 2023/04/04. We had added "Articles from 2017 January 1st to 2023 April 4th were searched." into 2.1.1 Databases and search terms.

5.       For QA the authors used the QUIPS AND RQS tool: What was the scoring of both the tools
==> Thanks for your suggestion. We have added the reference.

Secondly the authors need to put the filled-up tool for their study in the supplementary table as this needs to be reviewed.
==> We presented the QUIPS and RQS filled-up results in Figure 2 and Table 3.

6.       The have to define Overall survival also as they have defined response and PFS.
==> We added "The 6-month OS has defined whether the patient is alive within 6 months after treatment." into section 2.3. Definitions of 6-month progression-free survival and overall survival.

7.       In figure -1 Instead of records, the authors should mention studies/articles.
==> We had changed them into studies/articles.

8.       Heterogeneity Assessment Threshold Effect Assessment (HSROC) gives the sources of heterogeinity. I would recommend the authors to include a HSROC curve as it will add to the scientific rigour of the manuscript.
==> Though HSROC assessment is a good tool to present the sources of heterogeneity in diagnostic topics. However, due to our limited collected data and the ability software "Revman". We would add this to the limitation section.

9.       In the results section, the authors have used only one diagnostic metrics i.e is AUC. I understand that AUC is important and the primary diagnostic metric in a metaanalysis. However other diagnostic metrics like pooled specificity, pooled sensitivity should be calculated and should be represented in a forest plot as done for AUC.
==> Due to our limited collected data and the ability software "Revman", we could present the funnel plot for 6-month response AUC, PFS HR, and OS HR in the supplementary material. For pooled specificity and pooled sensitivity, we added them to our limitations.

10.    In table -2 the coloumn having clinical features is redundant as it has not been used for the analysis purpose, hence the coloumn of clinical features can be removed from the table 2.
==> We would like to reserve them while we mentioned this in 3.1.3 Radiomics and Image analysis:"Clinical features were incorporated into five studies[32-34, 36, 38]. A variety of formulas were used to calculate the difference in radiomic features across studies, incorporating both follow-up and pretreatment information. "

11.   The table 2 should include the radiomic features that were significant in each study for the given endpoints
==> We would like to do this. However, the detailed radiomic features were not presented in all the studies. Therefore, we listed them in the supplementary Table S1.

12.   A funnels plot for PFS, OS should be plotted for publication bias depicting the asymmetry of the funnel plot. The authors should also apply the eggers test for statistical significance for any publication bias.
==> We have done the funnel plots and Egger's test in the new Figure4.

13.   Discussion section

a. The discussion should start with endpoint in question and why this metaanalysis / systematic review is being done.
==> We added a description of the aim of our study as the first sentence in the discussion section.

b. The authors should discuss about the strength and limitations of the 10 studies.
==> The strengths and limitations of the 10 studies were discussed in several parts in the discussion section, especially the 4.5. Future direction part. Since we focused on the biological interpretation and prognostic ability, we added a paragraph to focus on the radiomic features used in the 10 studies:
4.3. Comparison of radiomic features in the enrolled studies
While the 10 enrolled studies utilize radiomics/delta radiomics as predictive parameters, the radiomic features employed in the predictive models differ across these studies. To better understand the biological interpretation of radiomic features, we categorized radiomics into several groups based on IBSI[26]. Among the 6 studies that disclosed their predictive radiomic features, a majority of these features are texture-related, such as entropy[33, 35, 41], contrast[41], and grey level uniformity[34]. The findings regarding the importance of texture-based radiomics align with the heterogeneity observed in tumors, a factor proven to be associated with prognosis in solid tumors[46, 47]. Only in 2 studies did size-based radiomic features, such as tumor volume[33] and axis length[38], enter the final prediction model. These findings suggest that changes in texture may provide more information than the traditional volume change proposed by the Response Evaluation Criteria in Solid Tumours(RECIST) working group or its modified version for immunotherapy, iRECIST[48]. Furthermore, radiomics that captured both size and heterogeneity were also deemed significant. For instance, metrics such as large/small area low/high grey level emphasis[34, 35, 38], which address both size and heterogeneity, were employed in 3 studies included in our analysis.

c. If I am not mistaken, all the studies had different Immunotherapy agents, which would actually change the response rates and the OS. The authors are encouraged to describe this in detail in the manuscript and should explain how it might affect their study.
==> As we listed in Table 1, the immunotherapy agents differ and the details cannot be obtained from some studies. We would add this to our limitations.: "Finally, the immunotherapy agents varied across different studies, and detailed infor-mation could not be obtained from some studies. This limitation prevents us from evalu-ating the individual effects of immunotherapy."

c. The authors should describe in details, the strengths and the weakness of their study which is a very important aspect of the metaanalysis.
==> We added a section to the discussion:
4.4. Strength and limitations
The most significant strength of this study is its status as the first meta-analysis discussing the predictive power of delta radiomics. Both QUIPS and RQS assessments were conducted in this study. However, several limitations were encountered. Firstly, out of the 223 studies/articles searched, only 10 studies could be included in the final analysis, indicating that delta radiomics is a relatively novel predictive parameter. Secondly, limited data hindered our ability to perform pooled sensitivity, pooled specificity, and heterogene-ity assessment, as well as threshold effect assessment (HSROC). Therefore, we presented the funnel plots in Figure S1, which demonstrated a low risk of publication bias. Thirdly, the varying definitions of radiomics and the inclusion of different radiomic features in various publications made it challenging for us to conduct further analysis. Finally, the immunotherapy agents varied across different studies, and detailed information could not be obtained from some studies. This limitation prevents us from evaluating the individual effects of immunotherapy.

 d. since this manuscript deals with delta-radiomics, the authors are advised to also dwell on the radiomic features in each study in the result section and also to include it to expand the discussion sectionS
==> We listed the radiomic features in supplementary material Table S1, and further expanded the discussion section.

Round 2

Reviewer 3 Report

Comments and Suggestions for Authors

The authors need to edit the english and also should correct the typos in the manuscript

Comments on the Quality of English Language

The authors need to edit the english and also should correct the typos in the manuscript

Author Response

We sent the final version for English editing through the MDPI English Editing service.